# Case study of a long-lived Siberian summer cyclone that evolved from a heat low into an Arctic cyclone

Franziska Schnyder<sup>1</sup>, Ming Hon Franco Lee<sup>1</sup>, and Heini Wernli<sup>1</sup>

<sup>1</sup>Institute for Atmospheric and Climate Science, ETH Zurich, Zurich, Switzerland

**Correspondence:** Franziska Schnyder (franziska.schnyder@usys.ethz.ch)

**Abstract.** Extratropical cyclones are known for strongly influencing mid-latitude weather in particular during the cold season and for their association with high-impact weather such as destructive winds and heavy precipitation. Cyclones occur typically in the oceanic storm track regions, and most studies about cyclone dynamics focused on cyclones that developed over the ocean. In this study, we investigate a particularly long-lived example of the lesser known Siberian summer cyclones. Based on a climatological analysis of Siberian summer cyclone tracks in ERA5 reanalyses during the period 1979–2021, we identify 9 events that are initially identified as typical heat lows. While there is a large variability in surface cyclogenesis conditions of Siberian summer cyclones, the Siberian heat lows form in very dry and hot environments and exhibit deep, convectively well-mixed boundary layers at genesis. In a detailed case study of a long-lived Siberian summer cyclone in July 2021, we show how the cyclone forms as a heat low during a heat wave in Kazakhstan. The cyclone then interacts with an upper-level trough, propagates across the Asian continent and evolves into an Arctic cyclone, which experiences rapid intensification and produces a warm conveyor belt with a poleward outflow approaching the North Pole and leads to the formation of a tropospheric potential vorticity cutoff in the Arctic. This case is unusual since subtropical heat lows are not known to propagate far from their location of origin. This unusual cyclone has a track length of almost 4000 km and it is associated with a heatwave initially, heavy precipitation during intensification, and an important upper-level flow anomaly in the Arctic. Comparison with the other Siberian heat lows shows that a similar development can be observed for the other cases, although not as pronounced and long-lived. This extraordinary case study also indicates how compounding high-impact events in different locations may be related to one single weather system.

Copyright statement. TEXT

#### 1 Introduction

Research on extratropical cyclones is a central and long-standing element of atmospheric dynamics, not least due to their substantial influence on midlatitude weather. In extreme cases, cyclones have the potential to generate destructive winds (e.g., Browning, 2004; Booth et al., 2015; Hirata, 2021), e.g., the storm "Lothar" in 1999 (Wernli et al., 2002), and heavy precipitation events (e.g., Yamamoto, 2012; Pfahl and Wernli, 2012; Barlow et al., 2019; Mohr et al., 2023), e.g., the Danube flooding in 2013

(Grams et al., 2014), or heavy snow fall in the eastern U.S. associated with the President's Day cyclone (Uccellini et al., 1985). Considering the dynamical effects of extratropical cyclones, they can significantly impact Rossby wave dynamics (Grams et al., 2011; Gray et al., 2014) and the formation and maintenance of blocks (Steinfeld and Pfahl, 2019), as a consequence of their associated warm conveyor belts (WCBs) injecting air with low potential vorticity (PV) into the upper troposphere. In some cases, this low-PV outflow of extratropical cyclones can contribute to the formation of Arctic anticyclones, with implications for summer-time sea ice melting (Wernli and Papritz, 2018).

30

With the aid of cyclone identification and tracking algorithms, global climatologies of cyclone activity including their annual cycle were investigated (e.g., Whittaker and Horn, 1984; Sinclair, 1997; Wernli and Schwierz, 2006; Hoskins and Hodges, 2019a, b). The climatology by Wernli and Schwierz (2006), which was obtained by applying a tracking algorithm to sea level pressure (SLP) minima, showed that although cyclone frequencies in the typical oceanic storm track regions are decreased in summer compared to winter, the opposite occurs in Siberia, leading to a maximum of cyclone activity in Siberia in summer. A different approach of tracking was applied by Hoskins and Hodges (2019a, b), who tracked maxima or minima of fields that were previously high pass-filtered. The tracking of vorticity maxima yielded a noticeably similar result as found by Wernli and Schwierz (2006), with a local maximum in summer near Siberia (Hoskins and Hodges, 2019a, their Fig. 1c).

Increased cyclogenesis frequency along the Siberian coast and in central Siberia has also been shown to contribute to the enhanced cyclone activity over the Arctic ocean during the summer months (Serreze et al., 1993; Serreze, 1995; Serreze and Barrett, 2008). Arctic cyclones are commonly defined as cyclones that exist roughly north of 65°N, irrespective of their place of origin (Serreze et al., 1993; Zhang et al., 2004; Sepp and Jaagus, 2011). The increased cyclogenesis frequency along the Arctic coast that contributes to the summer maximum in Arctic cyclones is likely explained by the development of a strongly baroclinic zone along the northern Eurasian coast due to strong surface heating gradients between sea ice and the continent in summer (e.g., Serreze et al., 1993; Serreze, 1995). But other Arctic cyclones have a mid-latitude origin over Siberia and propagate over large distances into the Arctic (Serreze and Barrett, 2008, their Fig. 4). As a final motivation to investigate Siberian cyclones in more detail, it is mentioned that mean track densities as well as mean cyclone intensities are expected to increase along the Siberian coast under the influence of climate change as a result of enhanced baroclinicity in these regions (Orsolini and Sorteberg, 2009). At the same time, cyclone activity in the North Atlantic and North Pacific storm tracks are projected to slightly decrease in summer (Orsolini and Sorteberg, 2009; Priestley and Catto, 2022). As a consequence, the relative contribution of Siberian cyclones to the entirety of cyclones is projected to increase.

While there have been several studies investigating climatologies of cyclone activity on a global scale, the focus of research about the detailed dynamics of individual systems and the environments in which they develop has predominantly been on cyclones that occurred in the primary cyclone regions, namely the North Atlantic and North Pacific storm tracks during winter (e.g., Joly et al., 1997; Rossa et al., 2000; Liberato et al., 2011; Zhao et al., 2020; Volonté et al., 2024a, b). Despite the prominent role of Siberian cyclones for the overall summer cyclone climatology and their relevance to Arctic cyclones, fairly little is known about their dynamics. Hence, in this study we first provide a general overview on Siberian summer cyclones and the environments in which they form. As the main part of our investigation, we present a detailed case study of a special type of Siberian summer cyclone, which started as a heat low and later transformed into an Arctic cyclone. Also, as shown

below, the selected cyclone was associated with different types of high-impact weather along its unusual track. To the best of our knowledge, such cyclones have not yet been reported in the literature. The next paragraph provides some background about heat lows, before then ending the introduction with an overview on the structure of the paper.

Heat lows are thermally driven cyclones that form over arid land masses, mainly in the subtropics, and, in certain regions, are prominent climatological features in the warmer months of the year. Examples include the Arabian heat low (Fonseca et al., 2022), the Australian heat low (Lavender, 2017), the Pak-India heat low (Bollasina and Nigam, 2011), and the West-African heat low (Lavaysse et al., 2009). These depressions in the SLP field arise from intense solar radiation leading to broad-scale ascent of near-surface air, dry convection and convergence at the surface. For this mechanism, a horizontal gradient in diabatic heating is essential. Due to their dependence on incoming solar radiation, these systems may exhibit a strong diurnal cycle (Portela and Castro, 1996). Simulations of Rácz and Smith (1999) investigating the dynamics of heat lows showed that while minimum SLP is reached in the late afternoon to early evening, the maximum relative vorticity at 100 m above ground is strongest in the early morning as a consequence of prolonged low-level convergence during the night. They further found that in spite of their cyclonic vortex at the surface, heat lows are associated with anticyclonic potential vorticity anomalies due to a greatly reduced static stability in the convectively well-mixed planetary boundary layer (PBL). Spengler and Smith (2008) extended the model by Rácz and Smith (1999) to include a representation of radiative heating and increased the horizontal resolution. They were then able to show the development of an in-up-and-out circulation during the mature stage of a heat low, which is similar to the one of a tropical cyclone. Above the PBL, the diverging outflow generates an anticyclone in the mid-troposphere directly above the shallow surface cyclone. These thermally driven cyclones are typically quasi-stationary. An exception to this can be found in Preissler et al. (2002), who briefly mentioned the presence of a mobile heat low passing Alice Springs in Central Australia.

The next section introduces the cyclone database and specific methods used for our analyses. In Sect. 3, an overview is given on cyclogenesis conditions of Siberian summer cyclones and about the selection of those cyclones that initially can be classified as heat lows and later propagate poleward. Section 4 then provides the detailed case study of a Siberian heat low in July 2021, which propagated across Siberia into the Arctic. We will discuss the development of this cyclone, the associated high-impact weather, and the dynamical processes that enabled this unusual evolution. For comparison, selected analyses for other Siberian heat lows are presented in Sect. 5, before ending with the main conclusions.

#### 85 2 Methods

90

#### 2.1 ERA5 cyclone tracks

Cyclones are identified in the global reanalysis data set ERA5 (Hersbach et al., 2020) from the European Centre for Medium-Range Weather Forecasts (ECMWF). Hourly fields of mean SLP, surface precipitation, potential temperature ( $\theta$ ), and potential vorticity (PV) were available, interpolated to a spatial grid of 0.5° by 0.5°.

The cyclones that are investigated and presented in this paper were identified using the feature-based tracking algorithm developed by Wernli and Schwierz (2006) and further refined by Sprenger et al. (2017). The algorithm first identifies a local

minimum in the SLP field as a cyclone centre. The algorithm then iteratively searches for the outermost contour that still encloses the cyclone centre but does not exceed a maximum length. This contour line then encloses the two-dimensional cyclone object. Cyclones are identified in this way by running the algorithm over the entire ERA5 data set. The subsequent cyclone tracking then constructs a plausible cyclone track by connecting the position of cyclone centres in time, using a first guess of cyclone motion based on recent propagation and suitable threshold criteria. For more details about the tracking, see Sprenger et al. (2017, their supplement). Additionally, because of the availability of hourly fields in ERA5, we apply a criterion for a maximum translation distance of 200 km between two hourly time steps (as opposed to 1000 km in the original tracking algorithm, which was developed for 6-hourly data) to avoid spuriously large jumps in the cyclone tracks over complex topography with noisy SLP fields.

In this study, we consider cyclones that occurred in the time period 1979–2021 and spent the majority of their lifetime (at least 75% of the time steps) over Siberia, our area of interest. This area was pragmatically defined with two rectangular areas: the first extending from 50°N to 75°N in the meridional and 60°E to 120°E in the zonal direction, respectively, and the second extending from 60°N to 75°N and from 120°E to 180°E. This domain is shown in the inset in Fig. 1. By implementing the criterion that the cyclones must spend at least 75% of their lifetime within the two boxes, cyclones originating within the boxes but spending the majority of their lifetime in the North Pacific storm track are effectively discarded. Although, the threshold of 75% was chosen pragmatically, additional analysis of the locations of genesis, maximum depth as well as lysis of the selected cyclones confirmed that this threshold successfully eliminates cyclones moving into the North Pacific storm track and that the selected cyclones indeed represent Siberian cyclones. Additionally, we impose a minimum lifetime criterion of 24 h as it has been done in previous papers (e.g., Sprenger et al., 2017; Heitmann et al., 2024). Since this study focuses on summer cyclones, only those that experience cyclogenesis in the months June, July and August (JJA) are considered.

# 2.2 Climatology of surface conditions at cyclogenesis

100

105

110

In order to determine the surface conditions for every identified cyclone at the time of cyclogenesis, several physical quantities are averaged within a radius of 100 km around the cyclone centre and over a range of vertical model levels. More specifically, we considered  $\theta$ , relative humidity, and PV. The radius of 100 km was chosen to represent the environment near the cyclone centre without being sensitive to the exact location of cyclogenesis. In comparison to vertically averaging between two fixed pressure levels, the choice of a range of model levels allows us to include cyclones in regions with varying topography. The lowest 30 model levels, which correspond to a height of about 3 km above ground, are chosen and thus give a representative indication of PV in the lower troposphere. Considering such a relatively tall layer is especially convenient since low-level PV anomalies may occur at different height levels. For  $\theta$  and relative humidity we decided to average over the lowest 10 model levels only, which corresponds to a height of about 300 m above ground and therefore represents the surface layer values. Sensitivity studies with a doubled radius or a different number of vertical levels (not shown) indicated that the results based on the choices explained above are robust.

#### 2.3 Warm conveyor belt masks

140

155

Additionally, WCBs are considered to investigate potential effects of these moist airstreams on Siberian cyclones, and their associated upper-level flow evolution. As in several previous studies (e.g., Madonna et al., 2014), WCBs are identified in ERA5 as trajectories that ascend rapidly in the vicinity of extratropical cyclones. Trajectories are started globally in the lower troposphere every 6 h from a set of starting points with a horizontal spacing of 80 km and a vertical spacing of 20 hPa between 1050 to 790 hPa. WCB trajectories are then selected based on the criteria of a minimum ascent rate of 600 hPa within 48 h. This study makes use of the so-called WCB masks, which were derived from the WCB trajectories as described by Heitmann et al. (2024). The WCB masks correspond to 6-hourly horizontal footprints of all WCB trajectories that are in a predefined pressure (*p*) range at the considered time. Three layers are considered and referred to as the WCB inflow (*p* > 800 hPa), WCB ascent (500 hPa < *p* < 800 hPa), and WCB outflow (*p* < 400 hPa). More details are given in Heitmann et al. (2024, their Sect. 2.2.2).

## 3 Overview of Siberian summer cyclones and Siberian heat lows

An overview of the surface conditions at cyclogenesis of all identified Siberian summer cyclones is shown in Fig. 1. For every cyclone, we consider  $\theta$ , relative humidity, and PV in the low-level environment of the cyclogenesis location, as described in Sect. 2.2. The figure reveals a huge variability of environments from which Siberian cyclones develop, with  $\theta$  ranging from 270 K to over 310 K and relative humidity between 15 and 100%. Low-level PV values vary between slightly negative values up to 4 PVU.

The scatter plot shows a noticeable linear relationship between  $\theta$  and relative humidity. Cyclones that develop in environments with high  $\theta$  (and, typically, at lower latitudes, see figure inset) also experience a relatively dry environment. On the other hand, cyclones that originate from a colder environment (at higher latitudes) typically experience higher humidity up to 100%. These large differences in relative humidity at cyclogenesis are relevant as they show that for genesis in an environment with  $\theta > 300$  K, relative humidity hardly ever exceeds 80% and therefore latent heat release is unlikely to play a role for cyclogenesis. In contrast, in environments with  $\theta 

Figure 1. ERA5 climatology (1979–2021) of near surface conditions at the time of cyclogenesis for all identified Siberian summer cyclones. The scatter plot shows  $\theta$  versus relative humidity (averaged over the lowest 10 model levels), and the color of the symbols indicates low-tropospheric PV (averaged over the lowest 30 model levels). Siberian heat lows (in the upper left corner of the diagram) are denoted with a star. The map shows the cyclogenesis locations of all cyclones, and the color of the dots indicates  $\theta$  at the time of cyclogenesis. The considered area of interest is indicated by the red contours in the map (see Sect. 2.1).

For the rest of this study, we focus on a small subset of Siberian summer cyclones, which form under very hot and dry conditions (Fig. 1, indicated by the star symbols). As discussed in more detail below, their cyclogenesis conditions are comparable to those of subtropical heat lows, and we therefore term these cyclones Siberian heat lows. We use a pragmatic threshold of relative humidity 

**Figure 2.** Tracks of the Siberian heat lows and time evolution of minimum SLP. The labels correspond to the event numbers in Table 1. The tracks highlighted in color are discussed in more detail in this paper. Cyclone 9 (red) is the main focus of this study. The 2 black crosses along the pressure evolution and track of this cyclone indicate the time and position of the cyclone at the onset of rapid intensification and maximum intensity, respectively.

#### 4 Life cycle of the propagating Siberian heat low in July 2021

#### 4.1 Overview


The cyclone emerges at 08 UTC on 3 July 2021, i.e., at 14 h local time, in northern Kazakhstan at  $52.5^{\circ}$ N/66.5°E with a central SLP of 1000 hPa. During the subsequent almost 8.5 d, it propagates across Siberia until it reaches the Arctic ocean after 6 d, at 15 UTC on 9 July (Fig. 2). As the cyclone moves over the continent, the central SLP only decreases marginally (it deepens less than 10 hPa in the first 4 d, Fig. 3). Despite this weak intensification, from t = 26 to t = 70 h, the cyclone is associated with a first WCB, leading to a first peak in precipitation in the environment of the cyclone (Fig. 3). Note that the precipitation rates may appear low due to areal averaging around the cyclone centre within a radius of 500 km. More precise precipitation patterns will be discussed in the following subsections.

Rapid intensification sets in after more than 6 d when the cyclone approaches the Arctic ocean, moves first over open ocean and subsequently over Arctic sea ice (Fig. 3). Central SLP decreases by almost 20 hPa in 30 h, and, at 18 UTC on 10 July,

Figure 3. Overview on the life cycle of the Siberian cyclone with genesis at 08 UTC on 03 July 2021 (t = 0), showing the time evolution of minimum SLP (black contour, in hPa) and hourly precipitation averaged in a 500 km radius around the cyclone center (blue shading, in mm h<sup>-1</sup>). Grey hatching indicates time periods when WCB ascent is present, and the blue and red lines denote times when the cyclone reaches the Arctic ocean and the sea ice edge, respectively. The track of this cyclone is shown in the inlet of Fig. 2.

maximum intensity is reached with a minimum central SLP of 969.4 hPa at 79°N (Fig. 3). Thereafter, the central SLP increases rapidly until the cyclone finally decays at 83°N at 18 UTC on 11 July 2021. The cyclone is associated with a second WCB as it crosses the open Arctic ocean during rapid intensification.

In subsequent sections, a detailed examination of the eventful life cycle of this cyclone will be given. The life cycle is divided into four phases, in which the cyclone first develops as a heat low shortly after an intense heat wave in Kazakhstan (Sect. 4.2), propagates over the continent with an upper-level trough (Sect. 4.3), intensifies rapidly over the open ocean concurrent with the formation of a WCB (Sect. 4.4), and eventually contributes to the formation of a tropospheric cutoff near the North Pole after its decay (Sect. 4.5). These four phases are indicated by black arrows at the bottom of Fig. 3. Additionally, two videos are provided in the electronic supplement, displaying PV at 330 K and  $\theta$  at 850 hPa during the full life cycle of the cyclone at an hourly resolution.

**Figure 4.** Synoptic situation before and at the time of cyclogenesis, i.e., at (a-c) 08 UTC on 1 July ( $t = -48 \, \text{h}$ ), and (d-f) 08 UTC on 3 July 2021 ( $t = 0 \, \text{h}$ ). The cyclone center is always in the center of the panels, denoted by a black star. All panels show SLP (black lines, every 4 hPa). In addition, left panels show  $\theta$  at 850 hPa (colors, in K), middle panels show hourly precipitation (color, in mm h<sup>-1</sup>) and PV at 850 hPa (pink, red, and orange contours for 1, 2, and 4 PVU, respectively), and right panels show PV at 330 K (colors, in PVU) and wind speed at 250 hPa (bold black contours, every  $10 \, \text{m s}^{-1}$  starting at  $30 \, \text{m s}^{-1}$ ). The gray dashed line in (f) indicates the location of the vertical cross section in Fig. 5a.

## 4.2 Phase I: Cyclogenesis



Cyclogenesis of this cyclone occurs during a severe heat wave in Kazakhstan (Kazhydromet, 2021). The heat wave prevailed throughout the first days in July with a maximum temperature of  $46.5^{\circ}$ C recorded in central-south Kazakhstan (Kazhydromet, 2021). During the days leading up to cyclogenesis, an intense PV cutoff is present on 330 K north of the Black Sea, west of the location of cyclogenesis (Fig. 4c). This cutoff, which has remained stationary since its formation 5 d prior to cyclogenesis on 28 June, is likely responsible for the poleward advection of very warm air to the location of cyclogenesis (Fig. 4a). At 08 UTC on 1 July 2021,  $\theta$  at 850 hPa at the location of cyclogenesis exceeds already 305 K. At the same time, there is no sign of precipitation north of a low-level PV anomaly in the region of cyclogenesis (Fig. 4b). During the following 48 h, the height of the boundary layer in the region of cyclogenesis grows significantly during daytime (up to 600 hPa, not shown).

At the time of cyclogenesis, at 08 UTC on 3 July, the PV cutoff is still present at upper levels west of the cyclone. It became more elongated and weaker, but it came closer to the region of cyclogenesis (Fig. 4f). Baroclinic interaction between the upper-

Figure 5. Vertical cross section in W-E direction of PV (color),  $\theta$  (black contours, every 3 K) and relative humidity (white contours for 20, 80, and 90%) at (a) cyclogenesis, i.e., at 08 UTC on 3 July (t = 0 h), and (b) at maximum cyclone intensity, i.e., at 18 UTC on 10 July 2021 (t = 178 h). The cross sections are centered at the cyclone center at the respective time steps. The location of the sections is shown by gray dashed lines in Fig. 4f for (a) and Fig. 7l for (b).




level cutoff and the incipient surface cyclone likely played a role for the initial decrease in SLP. At low levels, a PV anomaly has developed north of the cyclone centre at about 55°N (Fig. 4e). A vertical cross section reveals that the boundary layer in the region of cyclogenesis reaches heights of 800-600 hPa as indicated by constant values of  $\theta$  (Fig. 5a). The location of cyclogenesis coincides with the deepest well-mixed layer up to  $600 \,\mathrm{hPa}$ . The cross section also highlights the high  $\theta$  values of about 312 K that prevail in the emerging cyclone centre and the very dry air at the surface with relative humidity lower than 20% (Fig. 4d, 5a). Moreover, the very small positive to negative PV values throughout the lower troposphere at the cyclone centre correspond to the very low static stability (Fig. 5a). The combined characteristics in the region of cyclogenesis (high  $\theta$ , low relative humidity, low static stability, deep boundary layer with dry convection) are typically observed for heat lows (Rácz and Smith, 1999). The development of the deep PBL, as observed at the time of genesis, is likely the effect of vigorous dry convection associated with the formation of the heat low. Idealised simulations of heat lows in their mature stage by Spengler and Smith (2008) resulted in similarly looking vertical cross sections, which also revealed a very high, convectively mixed boundary layer with constant  $\theta$  reaching 4 km above ground. This prompts us to refer to this cyclone in its initial phase as a fully established Siberian heat low. However, the presence of an upper-level PV anomaly in the vicinity of the cyclone is unusual for subtropical heat lows, given that heat lows are known to be characterized by an anticyclonic wind field in the midtroposphere as a consequence of their in-up-and-out circulation (Rácz and Smith, 1999). As we discuss in the next sections, another approaching upper-level PV anomaly was most likely decisive for transforming the initial heat low into a propagating baroclinic system and advecting it poleward to a region of high baroclinicity.

# 230 4.3 Phase II: Propagation across Siberia






As the cyclone starts to propagate towards the northeast at  $t=36\,\mathrm{h}$ , a prominent PV trough approaches from the northwest (Fig. 6c). Associated with this trough, colder air is advected towards the cyclone from the northwest, leading to enhanced baroclinicity north of the cyclone (Fig. 6a, d), which goes along with the formation of an intense upper-level jet (Fig. 6c, f). On the one hand, the increased baroclinicity allows for the transition of the cyclone from a relatively shallow heat low to a troposphere-spanning baroclinic system with a PV anomaly at upper levels and high baroclinicity at the surface. On the other hand, the baroclinicity enables the formation of a first WCB (shown schematically in Fig. 3) along with increased precipitation rates in the vicinity of the cyclone center (Fig. 6e).

In the following two days, the cyclone continues to propagate along this baroclinic zone, always staying east of the still very pronounced PV trough. The trough absorbs the remnants of the PV cutoff that was present at cyclogenesis entirely (Fig. 6c, f, i). At the surface, a frontal structure starts to develop with an extended warm and cold front visible at 12 UTC on 7 July (Fig. 6g). A faint frontal structure is also apparent in the surface precipitation field (Fig. 6h). Frontal precipitation here is notably weaker than the precipitation associated with the WCB two days earlier (Fig. 6e, h). Low-level PV anomalies near the cyclone centre are still weak at this time. Also note that the cyclone, which traveled already far to the northeast, hardly intensified and, in the SLP field, forms a large but unimpressive and shallow feature. At the same time the intense jet at upper levels with wind speeds greater than 50 m s<sup>-1</sup> elongates (Fig. 6i, m). The development of this jet is consistent with the baroclinicity at the surface and of the strong PV gradient at upper levels along the downstream flank of the trough.

On 8 July 2021, the cyclone's central SLP falls below 990 hPa (Fig. 6k). Low-level PV anomalies become more intense with values larger than 2 PVU (Fig. 6l), mainly along the very extended warm and bent-back front where precipitation occurs with intensities up to  $3 \text{ mm h}^{-1}$ .

#### 250 4.4 Phase III: Rapid intensification

When the cyclone approaches the Arctic coast at 12 UTC on 9 July, i.e., more than 6 d after genesis, rapid intensification sets in, which lasts for about 30 h (Fig. 3). At this time, the cyclone is below a broad upper-level ridge, with an intense PV trough to its west. Maximum PV values in the trough exceed 11 PVU on 330 K in a small filament about 600 km west of the surface cyclone (Fig. 7c). North of the cyclone is the intense jet at upper levels with wind speeds above 50 m s<sup>-1</sup> (Fig. 7c). This jet is consistent with the very large meridional PV gradient at 330 K. At the surface, the cyclone is still located in a strongly baroclinic zone due to extraordinarily warm temperatures along the Arctic coast (Fig. 7a). This baroclinicity enables the generation of a second WCB (shown schematically in Fig. 3, Fig. 8). Baroclinicity was additionally quantified by calculating the Eady growth rate (EGR) between 850 and 500 hPa as in Besson et al. (2021), and the additional analysis of EGR (not shown) confirmed the exceptional potential for baroclinic intensification in this phase of the cyclone's life cycle.

From 15 UTC on 9 July to 04 UTC on 10 July the cyclone propagates over the open ocean (Fig. 3). In July 2021 the sea ice extent was anomalously low, leaving the entire Siberian coast unusually ice free and allowing for an additional moisture source. As a result of intense condensation in the ascent region of the WCB, precipitation rates are high with values exceeding

Figure 6. Same as Fig. 4, but during the propagation phase of the cyclone at  $(\mathbf{a-c})$  20 UTC on 4 July  $(t=36\,\mathrm{h})$ ,  $(\mathbf{d-f})$  12 UTC on 5 July  $(t=52\,\mathrm{h})$ ,  $(\mathbf{g-i})$  12 UTC on 7 July  $(t=100\,\mathrm{h})$ , and  $(\mathbf{k-m})$  16 UTC on 8 July 2021  $(t=128\,\mathrm{h})$ . The slightly bolder SLP contours around the cyclone centre in  $(\mathbf{k-m})$  indicate 990 hPa.

Figure 7. Same as Fig. 4, but during the intensification phase of the cyclone at (a-c) 12 UTC on 9 July ( $t = 148 \,\mathrm{h}$ ), (d-f) 19 UTC on 9 July ( $t = 155 \,\mathrm{h}$ ), (g-i) 12 UTC on 10 July ( $t = 172 \,\mathrm{h}$ ), and (k-m) 18 UTC on 10 July 2021 ( $t = 178 \,\mathrm{h}$ ). The gray dashed line in Fig. 71 indicates the location of the vertical cross section in Fig. 5b, and the blue contour in the left panels indicates a sea ice concentration of 75%.

 $6 \,\mathrm{mm}\,h^{-1}$  near the cyclone centre (Fig. 7e, 8a). Simultaneously, latent heating leads to diabatic PV production that intensifies the low-level PV anomaly with PV > 4 PVU locally (Fig. 7e). The vertical coupling of the increasingly intense low-level PV anomaly and the high PV at upper levels exceeding 10 PVU in a westward tilting configuration are consistent with the rapid intensification of the cyclone in this phase.

Diabatic production of low-level PV persists while the cyclone moves over the sea ice covered ocean after 05 UTC on 10 July. Until 12 UTC on 10 July the low-level PV anomaly has grown in size considerably and covers the entire cyclone centre (Fig. 7h). At upper levels, the cyclone centre is located underneath high PV values of 11 PVU, as the trough mentioned before has caught up with the poleward propagating cyclone (Fig. 7i).

At 18 UTC on 10 July the cyclone reaches a fairly barotropic structure and maximum intensity with a minimum SLP of 969.4 hPa. Although the PV profile does not correspond to a coherent PV tower, the amplitude and spatial extent of the low-level and upper-level PV anomalies are impressive (Fig. 5b). High relative humidity in the lower troposphere suggests that the anomaly was likely produced due to condensation associated with ascent in the WCB. At the same time the baroclinicity in the vicinity of the cyclone core has decreased notably compared to 23 h earlier and the entire core of the cyclone has become significantly colder and engulfed by cold polar air (Fig. 7d, k).

# 4.5 Phase IV: Arctic cyclone and decay






After the rapid intensification, the cyclone steadily decays as it moves further into the Arctic region. Nevertheless, the impact by the second WCB to the upper troposphere remains prominent. The outflow of the WCB extends remarkably far poleward, whereby the northernmost WCB trajectories reach a latitude well beyond 85°N at 18 UTC on 10 July (Fig. 8a, c). Previous WCB climatologies indicated that WCBs over Siberia with an outflow over the high Arctic occur only rarely during the summer months (Eckhardt et al., 2004; Madonna et al., 2014; Heitmann et al., 2024). Moreover, the frequency of WCBs reaching latitudes higher than 80°N is exceedingly small (Madonna et al., 2014; Heitmann et al., 2024), making the extent of this WCB exceptional. By transporting low-PV air to pressure levels above 400 hPa at these latitudes, this WCB significantly modifies the upper-level PV distribution and an amplified ridge reaching 80°N develops on 10 July at 330 K (Fig. 7i, m, and Fig. 8c).

As the cyclone is decaying it merges with another developing Arctic cyclone. Meanwhile, Rossby wave breaking occurs and the prominent ridge (Fig. 8c) develops into a narrow tropospheric PV streamer, which ultimately breaks up leading to the formation of a tropospheric PV cutoff located directly over the North Pole on 13 July (Fig. 8d, and video of PV at 330 K in electronic supplement). Such tropospheric PV cutoffs were shown to be relevant for enhanced summertime Arctic sea ice melting due to enhanced net surface shortwave radiation (Wernli and Papritz, 2018). It is remarkable that such a low-PV upper-level feature can be produced by a cyclone that originated 10 days earlier as a heat low over Kazakhstan.

Figure 8. WCB trajectories and their implications during the cyclone's rapid intensification phase. (a) WCB trajectories that are in their ascent phase (500-800 hPa) at 18 UTC on 9 July ( $t = 154 \,\mathrm{h}$ ). (b) Precipitation rates (colour) and WCB ascent mask (red contour) at 18 UTC on 9 July ( $t = 154 \,\mathrm{h}$ ). (c) and (d) PV on 330 K at 18 UTC on 10 July ( $t = 178 \,\mathrm{h}$ ) and at 01 UTC on 13 July ( $t = 233 \,\mathrm{h}$ ), respectively. The WCB outflow mask (thick black line) at 18 UTC on 10 July is additionally shown in (c). All panels show SLP (black lines, every 4 hPa).

#### 5 A comparison with other Siberian heat lows



In order to put the cyclone of this case study in perspective, we compare selected aspects of the cyclone of July 2021 (cyclone 9 in Table 1) with other Siberian heat lows. For the sake of completeness, we show all other 8 heat lows, however we will highlight 3 cyclones specifically, to show the similarities and the variability among the 9 Siberian heat lows. These 3 cyclones are the heat lows from June 1985 (cyclone 2), July 2005 (cyclone 6), and July 2017 (cyclone 8). While the first appears to be an unspectacular cyclone, the second experiences very rapid intensification shortly after cyclogenesis with similar intensification rates to the heat low from 2021. Lastly, the third has a similarly long track as the heat low from July 2021, but never experiences rapid intensification. Through this comparison, we find similarities and differences regarding conditions at cyclogenesis and in

terms of the PV development during the cyclone's life cycle.






Due to the selection criteria of high temperatures and low relative humidity at the surface, surface conditions are similar for all 9 Siberian heat lows. The vertical cross sections and PBL heights during the time of genesis are also similar among almost all Siberian heat lows and resemble the case study from July 2021 (Fig. 5a, and Fig. 9). Especially remarkable about these cross sections are the vertical lines of constant  $\theta$ , e.g., as for cyclone 2 and cyclone 6 (Fig. 9b,f), which highlight the well-mixed conditions and warm temperatures in the heat low centre. Furthermore, high PBLs (typically up to 600-700 hPa) are a common feature among the 9 cases, with a particularly high PBL in cyclone 2 (Fig. 9b). As mentioned in Sect. 4.2, high PBLs result from intense dry convection and are thus a typical characteristic of heat lows. Finally, weakly positive to negative PV values throughout the lower troposphere are observed in all vertical cross sections (Fig. 9b), consistent with the heat low from July 2021 (Fig. 5a). Note again, that these vertical cross sections also compare well with the vertical profiles of subtropical heat lows as described by Rácz and Smith (1999) and Spengler and Smith (2008).

Cyclogenesis conditions in terms of upper-level PV appear not as similar as they do for the surface conditions. While some heat lows have an upper-level PV anomaly in their vicinity at the time of genesis, others do not (not shown). Yet, during the later development of the heat lows, all cases start to propagate along the eastern flank of a PV trough or PV streamer (Fig. 10). In some cases, this can be observed already in the early stages of the cyclones' life cycle, e.g., 8 h after genesis for cyclone 6 (Fig. 10f), whereas in other cases this occurs much later, e.g., 78 h after genesis for cyclone 8 (Fig. 10h). The differences in the development after genesis can be shown by considering cyclones 2, 6 and 8: The heat low from 1985 (cyclone 2) propagates northwards (Fig. 2), while staying east of a very pronounced PV streamer. Figure 10b shows the cyclone 56 h after cyclogenesis, after the cyclone has already reached its maximum intensity. Unlike the heat low discussed in detail in the previous section (cyclone 9), cyclone 2 decays before the PV streamer can catch up with the surface cyclone. The cyclone does not experience rapid intensification. The opposite is the case for the heat low from 2005 (cyclone 6). The upper-level trough, which is west of the heat low at cyclogenesis (Fig. 10f), rapidly approaches the surface cyclone and only a few hours after cyclogenesis the heat low is located underneath a very strong-upper level PV anomaly (not shown). Accordingly, the cyclone experiences strong intensification shortly after cyclogenesis. The development of this heat low is similar to the late phase of cyclone 9. Lastly, the heat low from 2017 (cyclone 8) does not show an upper-level PV anomaly at cyclogenesis. Interesting about this case is that the propagation of this heat low is initiated only when a PV trough approaches the heat low from the west (Fig. 10h). During its lifetime of 10 d, the heat low then continues to propagate east of this PV trough across the Asian continent, similar to cyclone 9 (Fig. 2). Although the cyclone is eventually overtaken by the upper-level PV anomaly, the cyclone shows no sign of strong intensification. Unlike cyclone 9, this cyclone propagates into a region of weak baroclinicity as it approaches the Siberian coast (confirmed by consideration of EGR, not shown), which was likely the decisive missing ingredient inhibiting rapid intensification. Consistent with the much weaker intensification, this heat low features both a less pronounced upper-level and low-level PV anomaly (not shown).

**Figure 9.** Vertical cross sections in W-E direction of PV (color), θ (black contours, every 3 K), and relative humidity (white contours for 20, 80, and 90%) at cyclogenesis for: (a) cyclone 1 at 12 UTC on 21 August 1983, (b) cyclone 2 at 14 UTC on 04 June 1985, (c) cyclone 3 at 12 UTC 04 August 1991, (d) cyclone 4 at 07 UTC on 12 June 2004, (e) cyclone 5 at 17 UTC on 12 August 2004, (f) cyclone 6 at 11 UTC on 12 July 2005, (g) cyclone 7 at 12 UTC on 18 July 2012, and (h) cyclone 8 at 11 UTC on 23 July 2017.

**Figure 10.** Upper-level PV (color) and wind speed (black contours, every  $10 \,\mathrm{m\,s^{-1}}$  starting at  $30 \,\mathrm{m\,s^{-1}}$ ) at a chosen time during the life cycle of: (a) cyclone 1 at 14 UTC on 21 August 1983 (2 h after cyclogenesis), (b) cyclone 2 at 22 UTC on 06 June 1985 (56 h after cyclogenesis), (c) cyclone 3 at 17 UTC 04 August 1991 (5 h after cyclogenesis), (d) cyclone 4 at 14 UTC in 12 June 2004 (7 h after cyclogenesis), (e) cyclone 5 at 08 UTC on 13 August 2004 (15 h after cyclogenesis), (f) cyclone 6 at 20 UTC on 12 July 2005 (9 h after cyclogenesis), (g) cyclone 7 at 23 UTC on 18 July 2012 (11 h after cyclogenesis, and (h) cyclone 8 at 17 UTC on 26 July 2017 (78 h after cyclogenesis). All panels are cyclone centred (the centres are additionally indicated with a star) and the faint black contours show SLP in every panel (990 hPa and 970 hPa contours are highlighted with bolder contours).

Despite the differences in PV development during the cyclones' life cycles, we observe that all 9 Siberian heat lows propagate east of an upper-level PV anomaly. Both the presence of an upper-level PV anomaly and the fact that these cyclones propagate are generally absent in subtropical heat lows. This suggests, that the interaction with an upper-level PV anomaly may be an important factor for the propagation of Siberian heat lows. The interaction with the approaching upper-level PV anomaly can be regarded as a transition from a heat low to a midlatitude baroclinic system, with some analogies to the extratropical transition of tropical cyclones, where the interaction with a midlatitude upper-level PV anomaly also plays a central role (Keller et al., 2019). Although the cyclones originate as heat lows, they develop into baroclinic systems by coupling with an upper-level PV anomaly to the west and as such are able to propagate away from their region of genesis. The PV anomalies assume a key role, as they help advect the heat lows towards regions of moderate-to-high baroclinicity, where (rapid) intensification of the cyclone can occur.

## 6 Conclusion

In this study we presented an overview of Siberian summer cyclones identified in ERA5 from 1979-2021 as well as a special subset of these, which we refer to as Siberian heat lows. Genesis of Siberian summer cyclones occurs under a large variability of surface conditions, ranging from wet and cold to dry and hot environments. Furthermore, we found a noticeable linear relationship between  $\theta$  and relative humidity at the surface at the time of genesis, indicating a difference in the role of latent heat release and diabatic low-level PV production for cyclogenesis in hot and cold environments.

Within the entirety of the Siberian summer cyclones we identified a small subset of 9 cyclones that form as heat lows under very hot and dry conditions, have a lifetime of at least 24 h, and propagate across Siberia. Genesis of these cyclones occurs during the afternoon or evening hours in mostly the same region around Kazakhstan. These heat lows exhibit many similarities to subtropical heat lows including similar surface conditions at cyclogenesis, the timing of cyclogenesis and a deep, convectively well-mixed PBL. Yet, unlike classical subtropical heat lows, the identified Siberian heat lows propagate away from their location of genesis and may undergo rapid intensification during their lifetime. Their development resembles the life cycle of baroclinic extratropical cyclones featuring the propagation along the eastern flank of an upper-level PV trough or PV streamer, and the formation of an intense low-level PV anomaly (likely related to latent heat release). This suggests that the Siberian heat lows originally form as heat lows, just like subtropical heat lows, and then undergo a "transition" to baroclinic systems in the presence of an upper-level positive PV anomaly.

The detailed case study of a Siberian heat low in July 2021 shows an impressive evolution of one of these rare cases and offers relevant insights into how different compounding events may be connected by one singular dynamic weather system. The cyclone first originates as a heat low during a heat wave in Kazakhstan, then propagates thousands of kilometers across the Asian continent to the Arctic, where it undergoes rapid intensification. At this time the cyclone produces a second WCB, which leads to locally heavy precipitation, the formation of an intense positive low-level PV anomaly, and an low-PV outflow

that almost reaches the North Pole and amplifies the ridge east of the cyclone. Only few days later, poleward Rossby wave breaking occurs, resulting in the formation of a tropospheric cutoff located directly at the North Pole.

Finally, we would like to note that due to the minimum lifetime criterion applied in this study, very short-lived heat lows are omitted here, including a heat low with genesis during a heat wave in early August 2021 (Hayasaka, 2021) at 62°N/130.5°E close to Yakutsk. The occurrence of this heat low is especially impressive, as Yakutsk is known to be one of the coldest cities in the world during wintertime. Both our case study from July 2021 (cyclone 9) and this short-duration heat low a few weeks later at high latitudes formed during periods of heat waves in the respective genesis locations. In consideration of the projected increase in heat wave frequency and intensity with climate change, this suggests that such occurrences may become more frequent and at increasingly higher latitudes, such as in the case of Yakutsk. Considering our analysis of the case study of July 2021 and its impacts far away from its location of origin, increasing heat waves may not only have implications locally but also more remotely. In conclusion, this case study, although representing a rare event in the climatology of the last decades, might be particularly relevant for two reasons: (i) it reveals the enormous variability of cyclone life cycles and their associated impacts, and complements the collection of scientifically valuable cyclone case studies, which in the past were mainly performed for events in the main storm track regions (see introduction), with a detailed analysis of a summertime event over Central Asia, and (ii) it portrays a type of cyclone event, which is likely to become more frequent in a future warmer climate.

*Video supplement.* Supplemental information related to this paper is available from the ETH research collection at https://doi.org/10.3929/ethz-b-000730684

# Appendix A





# 385 A1 Overview of all Siberian winter cyclones

Figure A1 shows the cyclogenesis environment for Siberian cyclones developing during the winter months December, January and February. The distribution differs strongly from the environmental conditions of the summer cyclones (Fig. 1) featuring a cone-shaped distribution. The low-level PV values mostly range between 1–4 PVU and are much higher than that of the summer cyclones. The cyclones with the highest low-level PV values occur in rather unsaturated environments, implying that dry processes may be more important for the high PV values as well as cyclogenesis. One possibly important factor to consider here could be radiative PV production related to the formation of very stable boundary layers above sea ice. Lastly, maximum values of relative humidity at temperatures below 273.15 K are confined by the saturation vapour pressure with respect to ice. The spring and autumn distributions are a combination of the ones for summer and winter (not shown).

Author contributions. FS performed the study and wrote the paper with feedback about the result and text from MHFL and HW.

**Figure A1.** ERA5 climatology (1979 - 2021) of near-surface conditions at the time of cyclogenesis for Siberian winter cyclones. The scatter plot shows  $\theta$ , relative humidity and the color indicates low-tropospheric PV (see Fig. 1 for details).

Competing interests. At least one of the (co-)authors is a member of the editorial board of Weather and Climate Dynamics.

Disclaimer. TEXT

Acknowledgements. We thank Gwendal Rivière and two anonymous reviewers for their constructive feedback. The authors are grateful to Michael Sprenger for his support with the ERA5 cyclone tracks. FS and MHFL acknowledge funding from the Schweizerischer Nationalfonds zur Förderung der Wissenschaftlichen Forschung (grant no. 209135) and from ETH Zurich (grant no. ETH-06 21-1), respectively.

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
