# Peer review of "Case study of a long-lived Siberian summer cyclone that evolved from a heat low into an Arctic cyclone"

_EGUsphere, 2025_

## Author Comment (AC1)

**Final author comments for egusphere-2025-1724**

**Case study of a long-lived Siberian summer cyclone that evolved from a heat low into an Arctic cyclone**

by Franziska Schnyder, Ming Hon Franco Lee, and Heini Wernli

23 July 2025

We are most grateful to the three reviewers for their detailed and constructive comments that help us to further improve the manuscript. Based on the reviewers' suggestions, we will implement several changes in the manuscript. The main changes are that:

- We will calculate the Eady growth rate as a measure for baroclinicity and investigate this field along the track of the cyclones to better interpret the role of baroclinicity for the cyclone evolutions.
- We test a few sensitivities to the choice of certain parameters (cyclone residence time over Siberia; vertical levels for calculating low-level TH, RH, and PV).
- We improve the readability of some of the figures by adding more latitude and longitude labels and subtitles along the axes.

This document presents the reviewers' comments in blue and our responses in black.

**Reviewer 1**

Comments:

1. *Line 108-116. How sensitive are the results to the specified radius of 100 km that represents the environment near the cyclone? The low-level PV is calculated in the lowest 30 levels. Would it not be more reasonable to select levels above or in the upper part of the PBL? Since RH and theta are averaged only up to 300 m, how sensitive are the results to this threshold?*

We thank the reviewer for this comment. We have not tested these sensitivities yet, but we plan to do a limited set of sensitivity tests during the paper revisions. About PV: we think it is important for this study to calculate low-level PV in the boundary layer, because this then nicely reveals one of the particular aspects of the heat lows (low static stability in the well-mixed boundary layer and therefore very low PV values).

2. *Lines 149-155: The "subjective threshold" (RH < 20%) for defining Siberian heat lows could be better justified. Can you provide more information on how this threshold was chosen. Can you provide more information on the sensitivity of the selected cases to this threshold. Could you use RH and PV together to better define the subset of cyclones?*

We agree that the choice of our threshold (RH < 20%) is subjective, simple, and ad-hoc. However, in our view, it fulfills the purpose of this study. Figure 1 shows the overall distribution of RH, TH, and PV at the time of cyclogenesis. The upper-left part of the statistical distribution clearly stands out: these few cyclones have very low RH, very high TH, and again very low PV. We think it is fair to say that these cyclones are "special", i.e., that it is interesting to investigate them separately. Several criteria could be chosen to identify these cyclones: for instance a combined criterion of RH < 20% and PV < 0.5 pvu would identify the same cyclones as with our simple criterion based on RH only. And relaxing the RH criterion a bit more, e.g., RH < 30% and PV < 0.5 pvu would identify only a few more cases. In the end we aimed for the roughly 10 most special cyclogenesis events in this phase space, and we think that our simple criterion is doing the job well enough. We will improve our formulation to make clear that this threshold is subjective and pragmatic, and that it serves to identifiy similar cases in the upper-left corner of the phase space.

3. *Table 1: The inclusion of 8 other Siberian heat lows provides a broader perspective, which is great. However, the discussion could benefit from a more quantitative comparison; perhaps including the PV and track length/maximum latitude in Table 1 would help.*

We thank the reviewer for this comment and will make the according changes in the manuscript.

4. *Section 4.2 and 4.3. What is the role of the rather deep PBL in the cyclogenesis stage. Are the low-level PV and theta diagnostics representative of the cyclone environment since the boundary layer is so deep?*

We would argue that the deep boundary layer is an effect of the dry convection associated with the formation of the heat lows. We will make sure to convey this message more clearly in Sect. 4.2. Since the vertical cross-sections show relatively uniform values of PV and TH in the deep convective boundary layers, we think that our simple diagnostics (averaging PV and TH across 30 and 10 levels, respectively) are meaningful and representative.

5. *Section 4.4. It would be worthwhile to quantify the role baroclinicity through the evolution, perhaps through a bulk Eady index, to better understand the rapid intensification period.*

We thank the reviewer for this comment. We will perform an additional anaylsis investigating the baroclinicity (Eady growth rate) involved with this case and the other 8 heat lows (as also suggested by the 2nd reviewer) to better understand the role of baroclinicity for the intensification of these cyclones.

6. *Lines 270-272. What are the important characteristics that allow the WCB to reach the high latitudes in this case?*

Thank you for this interesting question. Although investigating this question is rather challenging, we are quite confident that in this case, the extended warm air advection on the eastern flank of the upper-level PV anomaly (see left panels in Fig. 6 and 7) was key. The very high temperatures in parts of northern Siberia most likely went along with humid conditions (comparatively high values of specific humidity), which then, together with the pronounced upper-level forcing for ascent due to the approaching upper-level trough, provided the ingredients that are necessary for the formation of a WCB.

7. *Lines 295-298. Is there any significance to the regions of negative PV aloft in these cases? Is there any relationship between the high PBLs and buoyancy, and low stability in the early stage of cyclogenesis that might influence the very low values of PV aloft?*

The vertical cross-sections show regions with negative PV in the convective boundary layer, up to about 700 hPa. Is this what you mean by "PV aloft"? When it comes to the specific PV values in a convective boundary layer (e.g., of a heat low) then the accuracy of ERA5 data can be questioned because the details of the vertical structure in these regions most likely depend strongly on the boundary layer parameterization (and are poorly constrained by observations). Therefore, we would be careful in regarding PV values close to 0 pvu as fully reliable. However, we also don't think that the dynamics of the investigated cyclones depends in any way on the specific PV values in the boundary layer. In other words, we don't think that the regions with negative PV are dynamically relevant. In our understanding, the negative PV values mainly indicate that in these regions dry convection did not yet fully eliminate static instability.

Technical Comments:

1. *Abstract line 11: Perhaps rephrasing "a warm conveyor belt whose outflow almost reaches the North Pole" to "a warm conveyor belt with poleward outflow approaching the North Pole" would improve clarity.*

We thank the reviewer for this comment and make the according changes in the manuscript.

2. *Line 63: "Examples are the Arabian heat low..." could be clearer as "Examples include the Arabian heat low..."*

We thank the reviewer for this comment and make the according changes in the manuscript.

3. *Figure 1. The scatter plot could use some improvements, the inset map is a little small, for example, and the light blue and dark blue are not that distinct.*

We thank the reviewer for this comment and increase the size of the inset map in Fig. 1 by shifting it to the lower left corner.

4.  *Lines 230-231: "the precipitation that occurs here along the fronts is significantly weaker...". could be more clearer as: "Frontal precipitation here is notably weaker..."*

We thank the reviewer for this comment and make the according changes in the manuscript.

5.  *Line 250: Perhaps rephrasing "Concomitantly, diabatic production of PV intensifies..." to "Simultaneously, latent heating leads to diabatic PV production that intensifies..."*

We thank the reviewer for this comment and make the according changes in the manuscript.

6.  *Line 322: "are generally not reported for subtropical heat lows" could be clearer as: "are generally absent in subtropical heat lows.*

We thank the reviewer for this comment and make the according changes in the manuscript.

**Reviewer 2**

Main comments:

1)  *Figure 10 shows that all the 9 selected surface cyclones developed and propagated east of an upper-level PV anomaly but some of them deepened and some others did not. Such a westward tilt with height of the anomalies is a favourable configuration for baroclinic growth but does not seem to be sufficient. On line 318 an interpretation is provided: "the absence of a strong PV tower may explain why this cyclone does not intensify as strongly." Do you mean that the low-level PV anomaly is too shallow to baroclinically interact with the upper-level PV anomaly ? In my mind, an alternative explanation could be the difference in the strength of the baroclinicity. How are the mean temperature gradients or the Eady growth rate during the evolution of the surface cyclones ?*

We thank the reviewer for this question concerninng the role of baroclinicity, which was also raised by RC1. We will perform an additional analysis investigating the role of baroclinicity (Eady growth rate) for the intensification of the presented heat lows to clarify this point. Regarding the statement with the PV tower, we will improve the formulation.

2)  *About the position of the cyclone center. In several panels, the stars which are symbols indicating the cyclones centers are not located inside a close contour of SLP. Since the*

*cyclone center is identified as a SLP minimum, I do not understand this mismatch. In figs. 4b and 10f this is particularly obvious. In Fig.4e it seems that the SLP minimum is further north than the star and should be closer to the high PV.*

We thank the reviewer for this comment. Please note, that the algorithm used to identify the cyclone centres searches for SLP minima every 0.5 hPa, in the plot however we only show SLP contours every 4 hPa. Therefore, the algorithm may place the centre at a location where there is a small local minimum in the SLP field that does not show in the plots. Regarding Fig. 4b, we plot the synoptic situation before the cyclone first appears, and indicate with the star where the cyclone will have its genesis 48 h later. This is why the stars do not match the SLP minima. To avoid confusion, we will add information on the time relative to genesis in the figure caption to emphasise this better. Regarding Fig. 10f we will alternatively show the synoptic map 1 h later than in the first version (i.e., at 9 h after cyclogenesis), as then the center is more clearly visible within the minimum contour line.

Minor comments:

*Figure 1: maybe recall in the caption over which levels PV and potential temperature are averaged.*

We thank the reviewer for this suggestion and will add this information to the caption.

*Figure 1 / line 103 about the choice of the two geographical areas. How do you know that the most eastern box is not related to the initiation of cyclones moving into the North Pacific storm track? More discussions on the choice of the two boxes would be good.*

We thank the reviewer for this question. We require the cyclones to spend at least 75% of their track lifetime within the two boxes. A cyclone with a lifetime of 4 days could therefore spend 1 day over the North Pacific, while spending the other 3 days over Siberia. Thus, we would still consider this cyclone a Siberian cyclone. However, to check the sensitivity, we will do a test where we require that the cyclones spend 90% of their lifetime over Siberia.

*Figure 2: it would be good to show the time in days rather than in hours to more easily connect the time evolution of the cyclone of Figure 2 with the weather maps shown for cyclone 9.*

We thank the reviewer for this suggestion. Rather than showing the time in days, we will add information about the hours since genesis in the titles of Fig. 4, 5 and 6 (e.g. t = 48 h) as well as in the figure captions to facilitate connecting the plots.

*Line 178: please refer to Fig. 3 at the end of the sentence after "environment of the cyclone".*

Thank you for this suggestion, we will add a reference to Fig. 3.

*Figures 3 and 4 about the synoptic situation at the time of cyclogenesis. Figure 4f (3 July) suggests baroclinic interaction with the first PV anomaly that could explain the slight decrease in SLP during genesis. But the text says "this anomaly is most likely too remote to influence cyclogenesis". I am not entirely convinced by the statement because the distance between the two anomalies is about the size of the anomalies, roughly 1000 km, i.e the radius of deformation.*

Thank you, we agree with you interpretation of Fig. 4f and we will adjust our text and mention the likely baroclinic interaction of the upper-level PV feature with the incipient cyclone.

*Figure 7: please indicate more longitudes.*

We thank the reviewer for this comment and will add more longitude labels.

*Figure 8: please indicate more latitudes and more loingitudes.*

As above.

*Figures 5 and 9: please indicate subtitles for the x- and y-axes (pressure and longitude).*

Thank you for spotting this oversight; yes, we will add axes titles.

*Line 208: "the very small to negative PV values troughout the lower troposphere at the cyclone center correspond to the very low static stability".* It seems to me *that in Fig. 4e the cyclone center is more to the north than the star indicates and closer to the positive PV (see second main comment).*

We thank the reviewer for this comment. We would like to clarify here that the statement refers to the vertical cross section (Fig. 5a not Fig. 4e) and will also more clearly reference the according Figure in the text.

*Line 214: the text says that "the presence of the upper-level PV anomaly in the vicinity of the cyclone is unusual for subtropical heat lows". If I understood correcly, the text says an upper-level PV anomaly is present, it is unusual in the vicinity of the heat low but has no effect on the cyclone dynamics. And line 217 "another approaching upper-level PV anomaly was most likely decisive". Don't you think that the effect of an approaching upper-level PV anomaly on the surface cyclone depends on the environmental baroclinicity ? See main comment 1. It would be interesting to show the cyclone trajectory superimposed onto the mean low-level baroclinicity. My opinion is that an approaching upper level anomaly will help to advect the surface cyclone northward until it reaches a zone of strong baroclinicity (here the Arctic frontal zone) where the two anomalies can then interact with each oher. So maybe the first PV anomaly is useful to advect the surface cyclone further north and*

*has an impact on the track more than on the cyclone intensity because they are far from the main baroclinic zone at that time.*

We thank the reviewer for this interesting question. As stated in response to the Main Comment 1, we will have a better look at the role of baroclinicity (Eady growth rate) for the evolution of these heat lows. Also, we very much appreciate and agree with the following interpretation and we will include parts of this in the revised version of our paper: "My opinion is that an approaching upper level anomaly will help to advect the surface cyclone northward until it reaches a zone of strong baroclinicity (here the Arctic frontal zone) where the two anomalies can then interact with each oher. So maybe the first PV anomaly is useful to advect the surface cyclone further north and has an impact on the track more than on the cyclone intensity because they are far from the main baroclinic zone at that time."

*Line 276-277. I do not understand why the presence of PV cutoff enhances net surface shortwave radiation. Are you talking about anticyclonic cutoffs or cyclonic cutoffs ?*

We thank the reviewer for this comment. As stated in the manuscript we refer to a "tropospheric cutoff" meaning an anticylonic cutoff. This is typically associated with downwelling, clear-sky conditions and in effect enhanced net surface shortwave radiation.

*Line 308: "cyclone 2 decays before the PV streamer can catch up with the surface cyclone". Here again, I am a bit sceptical about such a statement. In Fig. 10b, I do see a surface cyclone to the west of the PV streamer, almost in phase quadrature (the edge of the PV streamer is above the SLP minimum). It is a classical feature of the beginning of rapid baroclinic growth. If cyclone 2 still decays with such a configuration, it means that baroclinicity is weak. Here again I suggest computation of the mean baroclinicity.*

We thank the reviewer for this comment and will investigate the role of baroclinicity in more detail.

**Reviewer 3**

1) *Figure 1: It would be good to list the domain coordinates of the two red polygons, or refer the reader back to the Section 2.1 text, to reiterate those specific areas and what they represent in the caption.*

We thank the reviewer for this comment. We will make the according changes to the maunscript.

2) *Figure 2: Consider putting number labels also in the map for ease of referencing events 1-5 in particular.*

Unfortunately, the tracks are rather similar (and for the "gray cyclones" relatively short). It is therefore difficult to properly distinguish and meaningfully label the tracks in the map. We think that the most prominent cyclones (events 6, 8 and 9) can be well identified.

3) *Line 199: Suggest change to "...already exceeds 305K."*

Thank you. We will make the according changes to the manuscript.

4) *Line 208: Do you mean "very small positive"? Please clarify.*

Yes we do mean "very small positive". We will make the according changes to the manuscript.

5) *Line 273: Do you have a sense this system had an impact on the early July Arctic sea ice conditions along/near its track (at least in the near term)?*

This is an interesting aspect; however, it is challenging to assess the impact of a single cyclone on sea ice. We had a look at monthly sea ice maps from Copernicus, e.g., https://climate.copernicus.eu/sea-ice-cover-july-2021, which indicate that the Arctic sea ice extent in July and August 2021 was unusually low, with a sea ice free ocean along the Siberian coast. The pattern of sea ice anomalies appears similar in the two months, making it difficult to attribute any changes in the sea ice to this one cyclone. However, the sea ice is an important factor to consider, as anomalously low sea ice along the Siberian coast may have contributed to stronger moisture availability and a poleward shift of the baroclinicity in this region. We will emphasize the unusually low sea ice extent in July 2021 in Sect. 4.4.

6) *Lines 321-322: Is it remarkable that heat lows propagate away from their genesis points or propagate nearly as far as the 2021 case in particular? Please clarify and consider adding a reference in support of the statement.*

We thank the reviewer for this comment. It is both remarkable that the heat lows propagate away from their genesis points and even more so that the cyclone in this case propagates so far. We will emphasize this even more (and we don't know of any other study that showed a similar behaviour).

---

## Author Response (AR1)

**Final replies for egusphere-2025-1724**

**Case study of a long-lived Siberian summer cyclone that evolved from a heat low into an Arctic cyclone**

by Franziska Schnyder, Ming Hon Franco Lee, and Heini Wernli

**11 August 2025**

We are most grateful to the three reviewers for their detailed and constructive comments that help us to further improve the manuscript. Based on the reviewers' suggestions, we implemented several changes in the manuscript. The main changes are that:

- We calculated the Eady growth rate as a measure for baroclinicity and investigated this field along the track of the cyclones to better interpret the role of baroclinicity for the cyclone evolutions.
- We tested a few sensitivities to the choice of certain parameters (cyclone residence time over Siberia; vertical levels for calculating low-level TH, RH, and PV).
- We improved the readability of some of the figures by adding more latitude and longitude labels and subtitles along the axes.

This document presents the reviewers' comments in blue and our responses in black.

**Reviewer 1**

**Comments:**

1. Line 108-116. How sensitive are the results to the specified radius of 100 km that represents the environment near the cyclone? The low-level PV is calculated in the lowest 30 levels. Would it not be more reasonable to select levels above or in the upper part of the PBL? Since RH and theta are averaged only up to 300 m, how sensitive are the results to this threshold?

We thank the reviewer for this comment. Sensitivity studies targeting the radius of averaging (100 km vs. 200 km radius) indicate that while there are small changes in the values of the environmental variables, the overall pattern in the TH-RH-PV phase space remains the same (compare Fig. R1, which is identical to Fig. 1 in the paper, and Fig. R2, which shows the sensitivity test with the larger radius). The same is found when comparing the choice of 10 levels vs. 15 levels for calculating TH and RH, and 25 vs. 30 levels for low level PV (not shown). Most importantly, the group of heat lows still stands out as cyclones that develop in very dry and hot environments. We included the sentence: "Sensitivity studies with a doubled radius or a different number of vertical levels (not shown) indicated that the results based on the choices explained above are robust." (line 120) to inform the reader of the robustness regarding the choice of threshold.

Figure R1: This figure corresponds to Fig. 1 in the paper. ERA5 climatology (1979–2021) of near surface conditions at the time of cyclogenesis for all identified Siberian summer cyclones. The scatter plot shows  $\theta$  versus relative humidity, and the color of the symbols indicates low-tropospheric PV averaged over a radius of 100 km around the cyclone centre. Siberian heat lows (in the upper left corner of the diagram) are denoted with a star. The map shows the cyclogenesis locations of all cyclones, and the color of the dots indicates  $\theta$  at the time of cyclogenesis. The considered area of interest is indicated by the red contours in the map.

Figure R2: As in Figure R1 but here the values are averaged over a radius of 200 km around the cyclone centre.

Lines 149-155: The "subjective threshold" (RH

Figure R3: Eady growth rate [day1] (colours) and PV at 330 K (red contours) during the rapid intensification of cyclone 9 at a) 12 UTC on 8 July 2021 (t=124 h), b) 00 UTC on 9 July 2021 (t=136 h), c) 12 UTC on 9 July 2021 (t=148 h), d) 19 UTC on 9 July 2021 (t=155 h), e) 12 UTC on 10 July 2021 (t=172 h), and f) 18 UTC on 10 July 2021 (t=178 h). The sea level pressure is additionally indicated in black contours and the cyclone centre with a black star.

We thank the reviewer for this comment. We investigated the role of baroclinicity (by looking at the Eady growth rate, EGR) to better understand the rapid intensification of the cyclone. (We calculated EGR as in Besson et al. (2021) based on vertical averages between 850 and 500 hPa.) Indeed, we find that EGR is large along the Siberian coast, when the cyclone approaches the coastline (Fig. R3b and c). As the cyclone propagates into the region of large baroclinicity north of the Siberian coast,

rapid intensification sets in (Fig. R3 d and e). Enhanced baroclinicity in this region may have contributed in 2 ways to the cyclone's rapid intensification: 1. Stronger baroclinic interaction with the upper level PV anomaly (dry-dynamically) and 2. Stronger diabatic production of PV at low levels due to stronger latent heating in a more rapidly ascending warm conveyor belt (moist-dynamically).

Besson, P., L. J. Fischer, S. Schemm, and M. Sprenger, 2021. A global analysis of the dry-dynamic forcing during cyclone growth and propagation. Weather Clim. Dynam., 2, 991–1009.

5. Lines 270-272. What are the important characteristics that allow the WCB to reach the high latitudes in this case?

Thank you for this interesting question. Although investigating this question is rather challenging, we are quite confident that in this case, the extended warm air advection on the eastern flank of the upper-level PV anomaly (see left panels in Fig. 6 and 7) was key. The very high temperatures in parts of northern Siberia most likely went along with humid conditions (comparatively high values of specific humidity), which then, together with the pronounced upper-level forcing for ascent due to the approaching upper-level trough, provided the ingredients that are necessary for the formation of a WCB.

6. Lines 295-298. Is there any significance to the regions of negative PV aloft in these cases? Is there any relationship between the high PBLs and buoyancy, and low stability in the early stage of cyclogenesis that might influence the very low values of PV aloft?

The vertical cross-sections show regions with negative PV in the convective boundary layer, up to about 700 hPa. Is this what you mean by "PV aloft"? When it comes to the specific PV values in a convective boundary layer (e.g., of a heat low) then the accuracy of ERA5 data can be questioned because the details of the vertical structure in these regions most likely depend strongly on the boundary layer parameterization (and are poorly constrained by observations). Therefore, we would be careful in regarding PV values close to 0 pvu as fully reliable. However, we also don't think that the dynamics of the investigated cyclones depends in any way on the specific PV values in the boundary layer. In other words, we don't think that the regions with negative PV are dynamically relevant. In our understanding, the negative PV values mainly indicate that in these regions dry convection did not yet fully eliminate static instability.

**Technical Comments:**

1. Abstract line 11: Perhaps rephrasing "a warm conveyor belt whose outflow almost reaches the North Pole" to "a warm conveyor belt with poleward outflow approaching the North Pole" would improve clarity.

We thank the reviewer for this comment and made the according changes in the manuscript.

2. Line 63: "Examples are the Arabian heat low..." could be clearer as "Examples include the Arabian heat low..."

We thank the reviewer for this comment and made the according changes in the manuscript.

3. Figure 1. The scatter plot could use some improvements, the inset map is a little small, for example, and the light blue and dark blue are not that distinct.

We thank the reviewer for this comment and increased the size of the inset map in Fig. 1 by shifting it to the lower left corner.

4. Lines 230-231: "the precipitation that occurs here along the fronts is significantly weaker...". could be more clearer as: "Frontal precipitation here is notably weaker..."

We thank the reviewer for this comment and made the according changes in the manuscript.

5. Line 250: Perhaps rephrasing "Concomitantly, diabatic production of PV intensifies..." to "Simultaneously, latent heating leads to diabatic PV production that intensifies..."

We thank the reviewer for this comment and made the according changes in the manuscript.

6. Line 322: "are generally not reported for subtropical heat lows" could be clearer as: "are generally absent in subtropical heat lows.

We thank the reviewer for this comment and made the according changes in the manuscript.

**Reviewer 2**

Main comments:

1) Figure 10 shows that all the 9 selected surface cyclones developed and propagated east of an upper-level PV anomaly but some of them deepened and some others did not. Such a westward tilt with height of the anomalies is a favourable configuration for baroclinic growth but does not seem to be sufficient. On line 318 an interpretation is provided: "the absence of a strong PV tower may explain why this cyclone does not intensify as strongly." Do you mean that the low-level PV anomaly is too shallow to baroclinically interact with the upper-level PV anomaly? In my mind, an alternative

explanation could be the difference in the strength of the baroclinicity. How are the mean temperature gradients or the Eady growth rate during the evolution of the surface cyclones?

Figure R4: As Fig. R3 but for cyclone 8 at a) 17 UTC on 27 July 2017 (t=102 h), b) 17 UTC on 28 July 2017 (t=126 h), c) 17 UTC on 29 July 2017 (t=150 h), d) 17 UTC on 30 July 2017 (t=174 h), e) 17 UTC on 31 July 2017 (t=198 h), and f) 17 UTC on 01 August 2017 (t=222 h).

Figure R5: As Fig. R3 but for cyclone 2 at a) 10 UTC on 5 June 1985 (t=21 h), b) 22 UTC on 5 June 1985 (t=33h), c) 10 UTC on 6 June 1985 (t=45 h), d) 22 UTC on 6 June 1985 (t=57 h), e) 10 UTC on 7 June 1985 (t=69 h), and f) 22 UTC on 7 June 1985 (t=81 h).

We thank the reviewer for this question concerning the role of baroclinicity, which was also raised by reviewer 1 (comment 4). We investigated the role of baroclinicity by looking at the EGR during the heat low evolutions. Compared to cyclone 9 (Fig. R3), where we find high EGR along the Siberian coast, we find rather weak baroclinicity in the environment of cyclone 8 as it approaches the Siberian coast (Fig. R4). This is in line with cyclone 9 intensifying rapidly as it is near the coast, whereas cyclone 8 does not. For cyclone 2, baroclinicity is surprisingly high immediately west of the cyclone's centre even as the cylcone starts to decay (Fig. R5). Although a zone of high EGR is very close to the cyclone centre, the cyclone seems to be unable to propagate into this zone of enhanced baroclinicity. We speculate that the blocking-like ridge north of the cyclone and the elongated PV streamer west of the cyclone inhibit the cyclone's

propagation towards the zone of high EGR, keeping it on a north-oriented track. As a consequence, the cyclone does not intensify much before it begins to decay.

Regarding our interretation of why cyclone 8 does not intensify rapidly, we improved our formulation to: "Unlike cyclone 9, this cyclone propagates into a region of weak baroclinicity as it approaches the Siberian coast, which was likely the decisive missing ingredient inhibiting rapid intensification (not shown)" (line 331-333).

2) About the position of the cyclone center. In several panels, the stars which are symbols indicating the cyclones centers are not located inside a close contour of SLP. Since the cyclone center is identified as a SLP minimum, I do not understand this mismatch. In figs. 4b and 10f this is particularly obvious. In Fig.4e it seems that the SLP minimum is further north than the star and should be closer to the high PV.

We thank the reviewer for this comment. Please note, that the algorithm used to identify the cyclone centres searches for SLP minima every 0.5 hPa; in the plot however we only show SLP contours every 4 hPa. Therefore, the algorithm may place the centre in a location where there is a small depression in the SLP field, that does not show in the plots. Regarding Fig. 4b, we show the synoptic situation before the cyclone first appears, and indicate with the star where the cyclone will have its genesis 48 h later. This is why the stars do not match the SLP minima. To avoid confusion, we added information on the time relative to genesis in the figure caption and the subplot titles to emphasise this better.

**Minor comments:**

Figure 1: maybe recall in the caption over which levels PV and potential temperature are averaged.

We thank the reviewer for this suggestion. We added this information to the caption.

Figure 1 / line 103 about the choice of the two geographical areas. How do you know that the most eastern box is not related to the initiation of cyclones moving into the North Pacific storm track? More discussions on the choice of the two boxes would be good.

We thank the reviewer for this question. We require the cyclones to spend at least 75% of their track lifetime within the two boxes. A cyclone with a lifetime of 4 days could therefore spend 1 day over the North Pacific, while spending the other 3 days over Siberia. Thus, we would still consider this cyclone a Siberian cyclone. By changing the threshold to 90%, the number of cyclones are reduced from 588 to 520, and we "lost" our case study. Yet, our case study cyclone spends the majority of it's lifetime over Siberia and rapidly intensifes along the Siberian coast. Thus, we argue that a threshold of 75% is sufficient, to identify Siberian cyclones. To clarify this, we added

the sentence: "By implementing the criterion that the cyclones must spend at least 75% of their lifetime within the two boxes, cyclones originating in the boxes but spending the majority of their lifetime in the North Pacific storm track are effectively discarded." (line 105-107)

Figure 2: it would be good to show the time in days rather than in hours to more easily connect the time evolution of the cyclone of Figure 2 with the weather maps shown for cyclone 9.

We thank the reviewer for this suggestion. Rather than showing the time in days, we added information about the hours since genesis in the titles of Fig. 4, 5, 6, 7 and 8 (e.g. t=-48 h) as well as in the figure captions to bring out the connection between the plots even clearer.

Line 178: please refer to Fig. 3 at the end of the sentence after "environment of the cyclone".

Thank you for this suggestion, we added a reference to Fig. 3.

Figures 3 and 4 about the synoptic situation at the time of cyclogenesis. Figure 4f (3 July) suggests baroclinic interaction with the first PV anomaly that could explain the slight decrease in SLP during genesis. But the text says "this anomaly is most likely too remote to influence cyclogenesis". I am not entirely convinced by the statement because the distance between the two anomalies is about the size of the anomalies, roughly 1000 km, i.e the radius of deformation.

Thank you, we agree with you interpretation of Fig. 4f and we adjusted our text and mention the likely baroclinic interaction of the upper-level PV feature with the incipient cyclone: "Baroclinic interaction between the upper-level cutoff and the incipient surface cyclone likely played a role for the initial decrease in SLP." (line 210-211

Figure 7: please indicate more longitudes.

We thank the reviewer for this comment. We added more longitude labels.

*Figure 8: please indicate more latitudes and more loingitudes.*

As above.

Figures 5 and 9: please indicate subtitles for the x- and y-axes (pressure and longitude).

Thank you for spotting this oversight; yes, we added axes titles.

Line 208: "the very small to negative PV values troughout the lower troposphere at the cyclone center correspond to the very low static stability". It seems to me that in Fig. 4e the

cyclone center is more to the north than the star indicates and closer to the positive PV (see second main comment).

We thank the reviewer for this comment. We would like to clarify here that the statement refers to the vertical cross section (Fig. 5a not Fig. 4e). We added a reference to Fig. 5a at the end of the sentence to make this point more clear. Regardig the location of the cyclone centre, please see our answer to main comment 2.

Line 214: the text says that "the presence of the upper-level PV anomaly in the vicinity of the cyclone is unusual for subtropical heat lows". If I understood correctly, the text says an upper-level PV anomaly is present, it is unusual in the vicinity of the heat low but has no effect on the cyclone dynamics. And line 217 "another approaching upper-level PV anomaly was most likely decisive". Don't you think that the effect of an approaching upper-level PV anomaly on the surface cyclone depends on the environmental baroclinicity? See main comment 1. It would be interesting to show the cyclone trajectory superimposed onto the mean low-level baroclinicity. My opinion is that an approaching upper level anomaly will help to advect the surface cyclone northward until it reaches a zone of strong baroclinicity (here the Arctic frontal zone) where the two anomalies can then interact with each oher. So maybe the first PV anomaly is useful to advect the surface cyclone further north and has an impact on the track more than on the cyclone intensity because they are far from the main baroclinic zone at that time.

We thank the reviewer for this interesting question. Our analysis indicated that in particular for cyclone 9 strong baroclinicity was present during the rapid intensification of the cyclone, thus agreeing with your interpretation: "My opinion is that an approaching upper level anomaly will help to advect the surface cyclone northward until it reaches a zone of strong baroclinicity (here the Arctic frontal zone) where the two anomalies can then interact with each oher. So maybe the first PV anomaly is useful to advect the surface cyclone further north and has an impact on the track more than on the cyclone intensity because they are far from the main baroclinic zone at that time." Indeed our additional analysis of the EGR during the intensification of cyclones 8 and 9 especially support this interpretation. We added this interpretation in the manuscript as follows:

"As we discuss in the next sections, another approaching upper-level PV anomaly was most likely decisive for transforming the initial heat low into a propagating baroclinic system and advecting it poleward to a region of high baroclinicity." (line 227 - 228)

"Unlike cyclone 9, this cyclone propagates into a region of weak baroclinicity as it approaches the Siberian coast, which was likely the decisive missing ingredient inhibiting rapid intensification (not shown)". (line 331-333).

"The PV anomalies then assume a key role, as they help advect the heat lows towards regions of high baroclinicity, where rapid intensification of the cyclone can occur." (line 343-345)

Line 276-277. I do not understand why the presence of PV cutoff enhances net surface shortwave radiation. Are you talking about anticyclonic cutoffs or cyclonic cutoffs?

We thank the reviewer for this comment. As stated in the manuscript we refer to a "tropospheric cutoff" meaning an anticylonic cutoff. This is typically associated with downwelling, clear-sky conditions and in effect enhanced net surface shortwave radiation.

Line 308: "cyclone 2 decays before the PV streamer can catch up with the surface cyclone". Here again, I am a bit sceptical about such a statement. In Fig. 10b, I do see a surface cyclone to the west of the PV streamer, almost in phase quadrature (the edge of the PV streamer is above the SLP minimum). It is a classical feature of the beginning of rapid baroclinic growth. If cyclone 2 still decays with such a configuration, it means that baroclinicity is weak. Here again I suggest computation of the mean baroclinicity.

We thank the reviewer for this comment. As previously described, we surprisingly find a zone of high baroclinicity in the vicinity of this cyclone (Fig. R5). Our interpretation is that this cyclone does not manage to propagate into the zone of high baroclinicity west of the cyclone centre, due to a strongly north-south oriented PV streamer, therefore inhibiting the rapid intensification of this cyclone.

**Reviewer 3**

1) Figure 1: It would be good to list the domain coordinates of the two red polygons, or refer the reader back to the Section 2.1 text, to reiterate those specific areas and what they represent in the caption.

We thank the reviewer for this comment. We added a reference to Sect. 2.1 in the figure caption to help the reader.

2) Figure 2: Consider putting number labels also in the map for ease of referencing events 1-5 in particular.

Unfortunately, the tracks are rather similar (and for the "gray cyclones" relatively short). It is therefore difficult to properly distinguish and meaningfully label the tracks in the map. We think that the most prominent cyclones (events 6, 8 and 9) can be well identified.

3) Line 199: Suggest change to "...already exceeds 305K."

Thank you. We changed the sentence as suggested.

4) Line 208: Do you mean "very small positive"? Please clarify.

Yes, we do mean "very small positive". We made the according changes to the manuscript.

5) Line 273: Do you have a sense this system had an impact on the early July Arctic sea ice conditions along/near its track (at least in the near term)?

This is an interesting aspect; however, it is challenging to assess the impact of a single cyclone on sea ice. We had a look at monthly sea ice maps from Copernicus, e.g., <a href="https://climate.copernicus.eu/sea-ice-cover-july-2021">https://climate.copernicus.eu/sea-ice-cover-july-2021</a>, which indicate that the sea ice extent in July and August 2021 was unusually low, with sea ice free ocean along the Siberian coast. The pattern of sea ice anomalies appears similar between the two months, making it difficult to attribute any anomalies in the sea ice to this one cyclone. To emphasise this we added the sentence: "In July 2021 the sea ice extent was anomalously low, leaving the entire Siberian coast unusually ice free and allowing for an additional moisture source" (line 260-262).

6) Lines 321-322: Is it remarkable that heat lows propagate away from their genesis points or propagate nearly as far as the 2021 case in particular? Please clarify and consider adding a reference in support of the statement.

We thank the reviewer for this comment. It is both remarkable that the heat lows propagate away from their genesis points and even more so that the cyclone in this case propagates so far. We will emphasize this even more (and we don't know of any other study that showed a similar behaviour).

---

## Author Response (AR2)

**2nd reply for egusphere-2025-1724**

**Case study of a long-lived Siberian summer cyclone that evolved from a heat low into an Arctic cyclone**

by Franziska Schnyder, Ming Hon Franco Lee, and Heini Wernli

22 September 2025

We are grateful to the reviewers for their additional constructive comments and are glad that our initial revisions have satisfied most of their main concerns. Based on the reviewer's comment, we improved the justification of the chosen threshold of 75% for the selection of our Siberian summer cyclones.

This document presents the reviewers' comments in blue and our responses in black.

**Reviewer 1**

I remain concerned about the generality of the choice of the two geographical boxes and the 75% lifetime threshold, particularly regarding contamination from the North Pacific storm track (another reviewer raised a similar comment). The authors' response is that changing the threshold to 90% removes their case study cyclone.

 The criterion seems to have been chosen to ensure the inclusion of a specific event, rather than being based on an objective definition of "Siberian cyclone." While the authors added a sentence to the manuscript explaining the choice, it does not resolve the underlying issue that the selection criteria might be arbitrary.

We thank the reviewer for this comment. We agree (as we did before) that our criteria are to a certain degree arbitrary, or as we prefer to put it, subjective and pragmatic. As far as we know there is no standard definition of "Siberian cyclones" and therefore we came up with a simple and pragmatic approach. We would like to make the point that such a degree of subjectivity pertains to all cyclone climatologies. Most of them, for instance, use a minimum lifetime criterion, which can be 1 to 3 days, where the latter clearly excludes many shorter-lived cyclones from the climatology. These choices are often not fully apparent to readers and users of climatologies, but they are always existing. Coming back to our approach, which asks for 75% of the cyclone track to occur in a "Siberian region", we still regard this choice as meaningful. In our view, it would not make sense to use a higher threshold, because we don't want to exclude cyclones (like our case study), which spend many days in Siberia, but happen to have a much shorter early or late phase of their lifecycle, which is outside the domain.

• The authors could provide a stronger, more objective justification. For example, they could show a plot of genesis locations and track densities for all cyclones to demonstrate that the 75% criterion effectively isolates a distinct population of cyclones that primarily evolve over Siberia, as opposed to those that quickly transition into the Pacific storm track.

We thank the reviewer for these suggestions. In addition to the locations of genesis (which is also shown in Fig.1 in the manuscript), we also show here – for all Siberian cyclones selected with our approach and the 75% threshold – the locations of maximum depth as well as the locations of lysis (Figs. R1, R2, R3). The figures clearly reveal that the selected cyclones are successfully confined to our area of interest, and few to no cyclones enter the North Pacific storm track. We added this information in the manuscript: "Although, the threshold of 75% was chosen pragmatically, additional analysis of the locations of genesis, maximum depth as well as lysis of the selected

Fig. R1: Genesis locations of the Siberian summer cyclones selected with our pragmatic approach. Blue dots show regular Siberian summer cyclones, orange stars indicate the genesis locations of the 9 Siberian heat lows. The box used to select the Siberian summer cyclones is shown in red.

Fig. R2: As Fig. R1 but for the locations of maximum cyclone depth.

cyclones confirmed that this threshold successfully eliminates cyclones moving into the North Pacific storm track and that the selected cyclones indeed represent Siberian cyclones".

Fig. R1: As Fig. R1 but for the locations of lysis.